# Machine Learning-Based Interpretable Modeling for Subjective Emotional Dynamics Sensing Using Facial EMG

**DOI:** 10.3390/s24051536

**Published:** 2024-02-27

**Authors:** Naoya Kawamura, Wataru Sato, Koh Shimokawa, Tomohiro Fujita, Yasutomo Kawanishi

**Affiliations:** 1Computational Cognitive Neuroscience Laboratory, Graduate School of Informatics, Kyoto University, Yoshida-Honmachi, Sakyo, Kyoto 606-8501, Japan; kawamura.naoya.44x@st.kyoto-u.ac.jp; 2Psychological Process Team, Guardian Robot Project, RIKEN, 2-2-2 Hikaridai, Seika-cho, Soraku-gun, Kyoto 619-0288, Japan; koh.shimokawa@riken.jp; 3Multimodal Data Recognition Research Team, Guardian Robot Project, RIKEN, 2-2-2 Hikaridai, Seika-cho, Soraku-gun, Kyoto 619-0288, Japan; tomohiro.fujita@riken.jp (T.F.); yasutomo.kawanishi@riken.jp (Y.K.)

**Keywords:** facial electromyography (EMG), long short-term memory (LSTM), random forest regression, SHapley Additive exPlanation (SHAP), valence

## Abstract

Understanding the association between subjective emotional experiences and physiological signals is of practical and theoretical significance. Previous psychophysiological studies have shown a linear relationship between dynamic emotional valence experiences and facial electromyography (EMG) activities. However, whether and how subjective emotional valence dynamics relate to facial EMG changes nonlinearly remains unknown. To investigate this issue, we re-analyzed the data of two previous studies that measured dynamic valence ratings and facial EMG of the corrugator supercilii and zygomatic major muscles from 50 participants who viewed emotional film clips. We employed multilinear regression analyses and two nonlinear machine learning (ML) models: random forest and long short-term memory. In cross-validation, these ML models outperformed linear regression in terms of the mean squared error and correlation coefficient. Interpretation of the random forest model using the SHapley Additive exPlanation tool revealed nonlinear and interactive associations between several EMG features and subjective valence dynamics. These findings suggest that nonlinear ML models can better fit the relationship between subjective emotional valence dynamics and facial EMG than conventional linear models and highlight a nonlinear and complex relationship. The findings encourage emotion sensing using facial EMG and offer insight into the subjective–physiological association.

## 1. Introduction

Analyzing the relationship between subjective emotional experiences and physiological signals is of practical and theoretical significance. From a practical perspective, physiological indices of subjective emotions are unbiased indicators of emotional states that enable the prediction of subsequent behaviors [1]. Theoretically, it has been posited [2,3,4] that activities encapsulate the core of subjective emotional experiences. If objective physiology and emotion align, it can offer valuable insight into understanding the psychological mechanisms underlying subjective emotional experiences.

Recent psychophysiological studies have revealed that changes in facial electromyographic (EMG) signals, recorded from corrugator supercilii (cEMG) and zygomatic major (zEMG) muscles, exhibit a linear relationship with subjective emotional valence dynamics [5,6,7]. The cEMG and zEMG muscles manifest emotion-related facial actions, such as lowering eyebrows and pulling lips. Valence, which can be positive, neutral, or negative, denotes the qualitative aspect of subjective emotional experiences within the dimensional framework of emotion [8,9]. In previous studies [5,6,7], the dynamic valence ratings and facial EMG activities of the cEMG and zEMG muscles were measured while participants viewed emotional film clips. The results of mixed effect modeling [5,6] and Pearson’s product-moment correlation [7] demonstrated negative and positive correlations of cEMG and zEMG, respectively, with dynamic valence ratings. In line with this data, a preceding study indicated that cEMG and zEMG responses to emotional films exhibited a negative correlation with dynamic valence ratings, although the latter dataset was derived from an independent sample [10]. While observing emotional pictures, another investigation reported positive correlations between continuous negative and positive value ratings and cEMG and zEMG activity, though the statistical significance of these correlations was not assessed [11]. These findings indicate that the facial EMG signal changes are linearly associated with subjective emotional valence dynamics.

However, it remains unknown whether the relationship between subjective emotional valence dynamics and facial EMG changes is nonlinear. Although polynomial regression (a simple method to evaluate nonlinear associations [10,11]) was used in a previous study [7] and indicated the superior performance of quadratic models over linear ones, more complex nonlinear associations were not thoroughly examined. Recently, machine learning (ML) methods, including deep learning models, have substantially advanced the nonlinear prediction of dynamic time series data. In the realm of research focused on estimating emotional responses through dimensional models, certain studies have used subject facial expressions captured on video as input data. These studies employ convolutional neural networks and long short-term memory (LSTM) to estimate temporal changes in valence and arousal, demonstrating a substantial improvement in accuracy compared with conventional models [12,13]. Several other studies have demonstrated that random forest (RF) regression (a combination of decision tree predictors [14]) can provide accurate predictions using continuous predictors with nonlinear association patterns and complex intrapredictor interactions [15,16,17]. Drawing from these findings, we hypothesize that ML methods, including RF and LSTM, may better estimate subjective emotion responses from physiological data than conventional methods. If substantiated, such advancements could enhance the methodology of emotion estimation, with potential applications in practical domains such as consumer behavior prediction and mental health monitoring.

A disadvantage of ML models is their black-box nature [14]. However, a novel method called SHapley Additive exPlanations (SHAP) [18] is known for interpreting ML models and is based on game theory. Studies have shown that SHAP tools can effectively quantify and visually elucidate nonlinear associations and complex interactions among predictors in ML models [16,17,19,20]. However, the applicability of these tools to sequence-dependent models such as LSTM remains debatable [21]. Therefore, we hypothesize that applying SHAP tools to RF models can offer insights into the association between subjective valence dynamics and facial EMG changes. The insight could have theoretical implications for the physiological underpinnings of subjective emotional experiences.

To investigate these hypotheses, we reanalyzed the data from two previous studies [5,7] that measured dynamic valence ratings, cEMG, and zEMG while participants viewed emotional film clips. In particular, we predicted second-by-second subjective valence ratings using 12 features extracted from 10-s segments of EMG data (e.g., mean and standard deviation (SD)) using multilinear regression, RF, and LSTM.

## 2. Materials and Methods

### 2.1. Data Acquisition

The data used in this study comprise continuous valence ratings and facial EMG data (cEMG and zEMG) collected from 50 participants obtained in two studies [5,7]. Figure 1 shows an example from the employed dataset. Both studies employed identical apparatus, stimuli, and procedures.

#### 2.1.1. Participants

Data were collected from 50 participants, comprising 24 females and 26 males, with an average age of 21.8 years and SD of 2.5 years. All participants, who were Japanese volunteers, provided informed consent after a comprehensive explanation of the experimental procedures. The ethics committee of Kyoto University approved this study.

#### 2.1.2. Apparatus

Stimuli were displayed using a Windows computer (HP Z200 SFF; Hewlett-Packard Japan, Tokyo, Japan), a 19-inch cathode ray tube monitor (HM903D-A; Iiyama, Tokyo, Japan), and presentation 14.9 software (Neurobehavioral Systems, Berkeley, CA, USA). An additional laptop running on Windows (CF-SV8; Panasonic, Tokyo, Japan) was used for continuous cued-recall ratings.

#### 2.1.3. Stimuli

Three films, created by Gross and Levenson [22], representing highly negative (angry), intermediately negative (sad), and neutral stimuli, were employed in this study. These films have been previously validated in a Japanese sample [23]. In addition, two other films were presented: one featuring a waterside scene with birds (representing contentment) and another portraying a comedic dialog between two individuals (representing high positivity or amusement). The average duration of the film stimulus presentation was 168.4 s, with an SD of 17.8 s (ranging from 150 to 196 s). Moreover, Gross and Levenson [22] developed a fear-inducing film for practical purposes. All stimuli were presented at a resolution of 640 horizontal × 480 vertical pixels, resulting in a visual angle of approximately $25.5^{\circ }$ horizontally and ×$11^{\circ }$ vertically.

#### 2.1.4. Procedure

The experiments took place in a soundproof, electrically shielded room on a one-on-one basis. Participants were briefed about the experiment’s objective to measure electrical skin responses while they watched the films. They were given about 10 min to become accustomed to the room where the experiment was conducted. After a practice film, five films for testing were shown in a pseudo-randomized order.

For each trial, a fixation point and a white screen appeared on the monitor for 1 and 10 s before each film was presented. Following this, another 10 s white screen was shown, and the Affect Grid [24], which graphically represented the two dimensions of valence and arousal using a nine-point scale, was presented. The participants were instructed to concentrate on the fixation point when it was visible, observe the film, and evaluate their subjective emotional response to each film by pressing keys. After providing their responses, the screen went black during the intertrial intervals, which were randomly set to 24 to 30 s. Physiological data were continuously recorded for all trials.

Upon the completion of all trials, each stimulus was displayed on the monitor two or more times. Concurrently, a laptop displayed horizontal and vertical nine-point scales for assessing valence and arousal, respectively. The participants were instructed to recall their subjective emotional experience during the initial viewing and continuously rate that experience in either the valence or arousal dimension by moving a mouse, and the coordinates were continuously recorded. The participants rated valence first, followed by arousal. This cued-recall procedure was employed to collect two types of continuous rating data (specifically, valence and arousal) that were challenging to assess simultaneously during the initial viewing. A previous study reported a positive correlation between cued-recall continuous ratings and online continuous ratings in response to film clips [5,25]). However, this study did not analyze overall and continuous valence and arousal ratings.

#### 2.1.5. Physiological Data Recording

EMG data were recorded from cEMG, zEMG, and trapezius muscles using sets of pre-gelled, self-adhesive 0.7 cm Ag/AgCl electrodes with 1.5 cm interelectrode spacing (Prokidai, Sagara, Japan). The electrodes were placed on the left side of the face, with the ground electrode positioned in the middle of the forehead. The data were amplified, filtered online (band pass: 20–400 Hz), and sampled at 1000 Hz using an EMG-025 amplifier (Harada Electronic Industry, Sapporo, Japan), a PowerLab 16/35 data acquisition system, and LabChart Pro v8.0 software (ADInstruments, Dunedin, New Zealand). A 20-Hz low-cut filter was applied to remove motion artifacts. Simultaneously, a digital webcam (HD1080P; Logicool, Tokyo, Japan) was used to record the video and unobtrusively monitor motion artifacts. This study does not report other physiological measures (e.g., electrodermal activity).

### 2.2. Preprocessing

Data preprocessing was conducted using Psychophysiological Analysis Software 3.3 (from the Computational Neuroscience Laboratory of the Salk Institute) and in-house programs implemented on MATLAB 2018 or 2021 (MathWorks, Natick, MA, USA). EMG data were sampled for 10 s in the trial’s pre-stimulus baseline (white screen presentation) and film stimulus presentation of each trial, rectified, baseline-corrected using the mean value over the pre-stimulus period, and averaged every second. A coder checked for motion artifacts, and data across all conditions were combined for each participant.

Data segmentation was performed to augment the training dataset. Because subjective ratings were sampled at a 1 Hz rate, a 1 s-wide moving window was applied to capture as much information as possible at each point. This segmentation process generated 38,850 consecutive 10 s segments of physiological signal data and subjective valence ratings. Subsequently, time domain features were computed for each segment. The obtained features consist of six types: mean, maximum, minimum, SD, kurtosis, and skewness. These features were inputs for the linear regression and RF models. Ten segments were concatenated to examine the relationships between each segment, and the feature values of these segments were used as input data for the LSTM model.

### 2.3. Models

Both linear and nonlinear models were employed for comparison with conventional linear analysis. The primary objective of this study was to identify a model capable of predicting changes in valence over a time series. Two nonlinear models, the RF and LSTM models, were employed to achieve this. These models, with the small samples mentioned in the introduction, were selected based on their performance and their ability to incorporate past information from time series data [12,13,26]. Other models were excluded from this study because of their inaccuracy in preliminary analyses. The linear regression function provided by *scikit-learn*, a Python library, was employed for the linear regression model. RF, an ML network [14,27], demonstrates superior performance compared with other ML networks, efficiently predicting large datasets, handling high-dimensional features, and evaluating the significance of individual input features. The *RandomForestRegressor*, implemented in the Python library *scikit-learn*, was used. The optimal parameters were determined through a grid search using 10 random participants as the evaluation dataset. Finally, the RF parameters incorporate 20 decision trees with the decision tree model holding a maximum depth of six and the other parameters holding their default values.

LSTM, a recurrent neural network, can learn long-term dependencies from data and is particularly suitable for time series forecasting. It employs a sigmoid function as the activation function for the last linear layer. The structure of the LSTM model used in this study is shown in Figure 2. The Leaky ReLU and sigmoid functions are used as the activation functions. The Pytorch library, a Python tool, was employed to construct the model. Regarding the parameters, the hidden layer had a size of 64, the dropout rate was 0.2, the number of epochs was set to 50, the learning rate was 0.001, and the batch size was 100. Adam is used for parameter optimization, and default parameters were used except for the learning rate. The model with the highest accuracy on the evaluation dataset was selected for further analysis.

### 2.4. SHAP

To evaluate feature importance in RF, we introduced SHAP by Lundberg and Lee [18], a method for the feature importance based on game-theoretically optimal Shapley values [28]. The Shapley value is a method to distribute the *payout* (in this case, the model’s prediction) among the *players* (the features) fairly. It evaluates all possible combinations of features, determining each feature’s contribution to the difference in the model’s prediction when included against when excluded.

In summary, the Shapley values for linear models $\phi_{i}$ given a sample *x* are computed as follows:(1)$$\begin{array}{cc} \phi_{i} & =\sum_{S\subseteq F\setminus \{i\}}\frac{|S|!(|F|-|S|-1)!}{|F|!}\left(f_{S\cup \{i\}}\left(x_{S\cup \{i\}}\right)-f_{S}\left(x_{S}\right)\right). \end{array}$$
where F represents the set of all features S is a subset of features excluding feature *i*, $f_{S}$ is the model trained with features in set S, and $x_{S}$ denotes the set of feature values in set S. This formula calculates the average contribution of a feature across all possible feature combinations.

The term $\frac{|S|!(|F|-|S|-1)!}{|F|!}$ is a weighting factor, determining the proportion of all permutations of features that include the current subset S. The expression $f_{S\cup \{i\}}\left(x_{S\cup \{i\}}\right)-f_{S}\left(x_{S}\right)$ calculates the marginal contribution of feature *i* when added to subset S. When adapting $\phi_{i}$ to other models, sampling approximations are applied to Equation (Equation 1) to estimate the effect of the omitted features by integrating over samples from the training dataset [29].

This computation considers all possible combinations of features, ensuring a fair and comprehensive assessment of each feature’s contribution. It is a powerful method for understanding model predictions, particularly in complex models where interactions between features may not be immediately apparent.

SHAP interaction values extend the concept of SHAP values by quantifying the importance of individual features and the interactions between pairs of features in an ML model. These values aid in understanding how combining different features impacts the model predictions, offering insights beyond those provided by individual SHAP values.

The SHAP interaction value $\phi_{i,j}$ for a pair of features *i* and *j* in a prediction is computed as follows:(2)$$\begin{array}{cc} \phi_{i,j} & =\sum_{S\subseteq F\setminus \{i,j\}}\frac{|S|!(|F|-|S|-2)!}{(|F|-1)!}\delta_{i,j}\left(S\right).\;\;\;\;\;\;\;\; \end{array}$$(3)$$\begin{array}{cc} \delta_{i,j}\left(S\right) & =\left(f_{S\cup \{i,j\}}\left(x_{S\cup \{i,j\}}\right)-f_{S\cup \{i\}}\left(x_{S\cup \{i\}}\right)-f_{S\cup \{j\}}\left(x_{S\cup \{j\}}\right)+f_{S}\left(x_{S}\right)\right). \end{array}$$

This captures the interaction between features *i* and *j* and proves especially valuable in intricate models where feature interactions can substantially impact the resulting output.

The SHAP library, simplifies this complex calculation, offering methods to compute SHAP values for various model types and visualize the results. Thus making this powerful technique accessible for practical ML applications.

### 2.5. Data Analysis

#### 2.5.1. Prediction Performance Analysis

The model evaluation indices include the mean squared error (MSE) and the correlation coefficient between the model predictions and the actual values. These evaluation indices, the MSE and correlation coefficient for each model’s prediction of the subjective ratings for each input, allowed for identifying performance differences among linear regression, RF, and LSTM. Leave-one-out cross-validation was employed in this study, where data from one participant served as the evaluation dataset, and the other participants were used as the training dataset to train models. Consequently, each model’s accuracy (MSE and correlation coefficient) was computed for each participant serving as the evaluation dataset, and the average value across all participants was used as the overall result. The indices were subjected to statistical analysis using JASP 0.14.1 software [30]. First, the correlation coefficient of each model was analyzed using a one-sample *t*-test, and they were compared with zero (two-tailed). Subsequent differences in MSE values and correlation coefficients between models were examined using repeated-measure one-way analyses of variance (ANOVA) with the models as factors, followed by multiple comparisons using the Bonferroni method (two-tailed). Additionally, considering that model accuracy is a crucial prerequisite for the SHAP analysis in the next section, we used the RF model with the highest accuracy from the results obtained from this section.

#### 2.5.2. SHAP Analysis

First, we computed SHAP values for each instance. SHAP [18], a method used for explaining the output of RFs, relies on the concept of Shapley values from the cooperative game theory. These values offer insight into how a model’s predictions change with the contribution of each feature.

Second, we calculated the mean of the absolute SHAP values associated with the obtained hyperquantities. This metric allowed us to assess the significance of each input feature.

Subsequently, we examined scatter plots that illustrate the relationship between the input features and their corresponding SHAP values for each instance. The visual representations provided insights into the interactions between the primary features and the generated SHAP values.

Furthermore, we computed SHAP interaction values encompassing all features to investigate feature interactions. These values were then presented as a two-dimensional heat map, showing the interplay and relative importance of various features in the dataset.

To better understand these interactions, we scrutinized scatter plots illustrating the relationship between the input features and SHAP interaction values, focusing on two features with the most notable interaction effects.

## 3. Results

### 3.1. Prediction Performance Analysis

Figure 3 depicts the MSE and the correlation coefficient between the actual and predicted valence values. Representative instances of the actual and predicted valence ratings are shown in Figure 4. First, one-sample t-tests revealed that all models exhibited correlation coefficients significantly different from zero (t(49)>9.89,p<0.001,d>0.19). Subsequently, ANOVA demonstrated significant differences in MSE values among the models ($F\left(2,98\right)=17.90,p<0.001,\eta_{p}^{2}=0.27$). Upon conducting multiple comparisons, it was evident that the MSE values for the RF and LSTM models were significantly lower than that of the linear regression model (t(98)>3.84,p<0.001). By contrast, no significant difference was observed between the RF and LSTM models (t(98)=2.05,p=0.130). Similarly, ANOVA highlighted significant differences in the correlation between actual and predicted valence ratings among the models ($F\left(2,98\right)=32.14,p<0.001,\eta_{p}^{2}=0.40$). Upon further investigation through multiple comparisons, it was established that the correlation coefficients for the two ML models were significantly higher than that of the linear regression model (t(98)>6.69,p<0.001), with no significant difference between the first two (t(98)=0.46,p=0.644).

### 3.2. SHAP Analysis

Figure 5 shows the absolute average SHAP value results representing feature importance. The experimental results reveal that the mean cEMG and zEMG values are the most important features. Furthermore, it is evident that the SD values of the EMG signals, particularly those of zEMG, play a substantial role in prediction.

The SHAP dependency plots depict representative relationships between the EMG features; the predicted valence dynamics are shown in Figure 6, which represents the relationship between each feature and its corresponding SHAP values. For instance, the top-left panel illustrates the relationship between the mean value of cEMG and its SHAP value, demonstrating how the value changes as the feature is input. The plots for the mean cEMG and zEMG values reveal negative and positive relationships with the predicted valence dynamics. However, the patterns displayed are nonlinear; instead, they exhibit sigmoid and step function patterns for the mean cEMG and zEMG values, respectively. Similarly, the plots of the cEMG and zEMG SD values show negative and positive relationships with the predicted valence dynamics following a step function pattern.

Figure 7 depicts a plot illustrating all individual feature mean absolute SHAP interaction values. This plot emphasizes the significance of the feature interactions within our dataset. Notably, the highest degree of importance is attributed to the SHAP interaction values between the mean cEMG and zEMG values.

Figure 8 depicts the relationship between SHAP interaction values related to the mean cEMG and zEMG values, which are of utmost importance, and their respective features. In this figure, a discernible trend indicates a decrease in valence as cEMG activity increases. In addition, there is a trend of increasing valence when cEMG activity is high, and zEMG activity is low.

## 4. Discussion

Our MSE and correlation coefficient results (Figure 3) demonstrate that the RF and LSTM outperformed the linear regression model in predicting emotional valence dynamics based on changes in facial EMG. Although the linear regression model exhibits a significant correlation between the predicted and actual valence values, in line with previous findings [5,6,7], the ML models indicate relatively superior correlations. The above results suggest that the EMG of facial expressions does not solely follow linear relationships with subjective emotion but instead exhibits nonlinear and interactive relationships. The results align with previous research, indicating that ML models outperform linear models in predicting two-dimensional emotions from facial expression videos, EEG, and the autonomic nervous system in regression tasks [12,13]. Nevertheless, studies have yet to explore this issue to analyze the dynamic relationships between emotional valence and facial EMG. To the best of our knowledge, this is the first study to establish that ML models outperform linear regression models in predicting subjective emotional valence dynamics.

The prediction performances of the RF and LSTM models were similar. These results deviate from those of previous studies that indicated better predictive capabilities of LSTM than the RF on time-series data [31]. However, these results agree with the findings of other investigations that reported comparable predictive performance between these two ML models [32,33]. We postulate that one contributing factor to this inconsistency may be the limited sample size and relatively uncomplicated model structure employed in this study [32].

Further, our analysis, complemented by interpretation methods employing SHAP values, sheds light on how black-box-like [14] ML models offer superior predictive capabilities for the relationship between subjective emotional valence dynamics and facial EMG. First, the average SHAP value plot (Figure 3) indicates that the mean cEMG and zEMG values were the most important predictive features. This agrees with previous research, which used linear analyses and identified these values as contributors to the prediction of subjective valence dynamics [5,6,7]. In addition, our findings revealed the significance of SD values in facial EMG signals, a factor not emphasized in previous studies. Specifically, our results indicate that a high SD, signifying frequent movement of cheek muscles, correlates with heightened subjective valence.

The SHAP dependency plot (Figure 6) indicates the associations between the mean cEMG and zEMG values and their corresponding SHAP values. These associations exhibit negative and positive trends, in line with previous analyses [5,6,7]. However, the profiles of the relationships are nonlinear. In particular, the mean cEMG value displays a sigmoidal relationship with valence. This phenomenon suggests that brow muscle movement intensity reaches a plateau at extreme values on both positive and negative sides, whereas it undergoes continuous changes in the intermediate stage. Similarly, the zEMG mean value revealed a step function relationship with valence, suggesting that cheek muscles undergo abrupt changes in response to varying levels of pleasantness. Comparable phenomena were observed for the SDs of the EMG signals, but a step-function relationship was observed between the SD and the valence of cEMG. This observation suggests that it is feasible to discern emotions based on the activity levels of cEMG. In summary, these findings highlight the nonlinear relationships between subjective emotional valence dynamics and facial EMG changes.

Our analyses of the SHAP interaction values (Figure 7) revealed that the mean absolute values of the SHAP interactions between the cEMG and zEMG mean values were the highest. The SHAP dependence plot for this interaction (Figure 8) demonstrates that when the zEMG mean value is high, cEMG exhibits a correlation similar to standard SHAP values. However, when the zEMG mean value is low, the correlation is opposite to the normal correlation. In addition to the correlation with the standard SHAP values, the nonlinear models may offer superior predictive capabilities compared with linear regression because they distinguish emotions based on intricate interrelationships. Our results offer new insights into the interactive relationships between facial EMG features and subjective emotion valence dynamics.

Our results that the facial EMG is more useful in predicting subjective valence dynamics than previously expected have practical implications. A previous consumer psychology study that only analyzed a linear component [34] underscored that zEMG signals outperformed electrodermal activity measurements in the sensitive prediction of purchase decisions. This observation suggests that facial EMG can offer valuable insights into customer emotional states and purchasing behaviors, making it a valuable tool in marketing research. In clinical contexts, another study [35] reported that facial EMG recordings from cEMG demonstrated superior sensitivity in distinguishing between stress and normal states compared with electrodermal signals. These findings indicate the relevance of facial EMG data in assessing mental health states, potentially complementing clinical evaluations reliant on interviews and behavioral observations. Combined with ML models, these facial EMG changes can provide valuable predictive information about subjective emotion valence dynamics and behaviors.

Our results, which demonstrate nonlinear and interactive relationships between facial EMG and emotion valence dynamics, also have theoretical implications. The association between subjective and physiological emotional responses has piqued the interest of psychology ever since James [2] proposed that physiological signals constitute the essence of subjective emotional experiences [3,4]. However, as Cannon [36] initially argued, several studies failed to identify a consistent one-to-one correspondence between physiological states and subjective experiences of emotional categories [37,38]. One of the reasons for this may be that the linear analyses employed in previous studies failed to capture the nonlinear and interactive associations between physiological activities and subjective emotional experiences. Previously, some researchers have proposed mechanistic models to examine the association between subjective and facial emotion responses. For instance, facial muscle activations have been indicated to offer feedback to the brain regions, such as the amygdala, in the form of emotional proprioceptive signals [39]. However, few models specify the functions of associations among facial muscle activity, mediators, and subjective emotion states. Our results indicate that the physiological–mechanistic models underlying emotional dynamics should account for nonlinear and interactive associations between facial and subjective emotional signals.

One informatic limitation of this study is the need for a complex model or algorithm. Enhancing the accuracy of valence prediction may be achieved by introducing a more suitable model that better interprets physiological signals. Future endeavors will encompass preprocessing, model enhancement, and exploration, and the integration of emotion prediction with other indicators, such as autonomic nervous system biometric data and image recognition. The techniques employed in this study will serve as a guide for developing more sophisticated emotion prediction models. A limitation of the SHAP analysis is that SHAP values are designed to reveal correlations between features and predictions rather than establish causality. Therefore, empirical experiments must be conducted to confirm causal relationships.

## 5. Conclusions

In conclusion, this study demonstrates that EMG can predict human emotions. It uses a regression model to estimate continuous valence values from facial EMG data. In particular, we compared linear regression with more advanced nonlinear models, such as the RF and LSTM models. The datasets used in this study included EMG data of facial expressions and subjective valence ratings from 50 participants who watched five videos. The results show that nonlinear models outperform linear regression in terms of accuracy. In addition, distinct correlations between cEMG and zEMG and their corresponding RF valence predictions suggest that the two EMG signals can help differentiate between increased and decreased valence predictions. The importance of this research is highlighted by its potential to advance techniques in both theoretical and practical areas, such as real-time emotion detection using facial muscles, enhancing emotion recognition techniques, and refining psychological theories on emotion dynamics. For future research, it is recommended that empirical experiments be conducted to validate this study’s results. To develop this promising field further, expanding the complexity of the models should be considered.

## Figures and Tables

**Figure 1 sensors-24-01536-f001:**
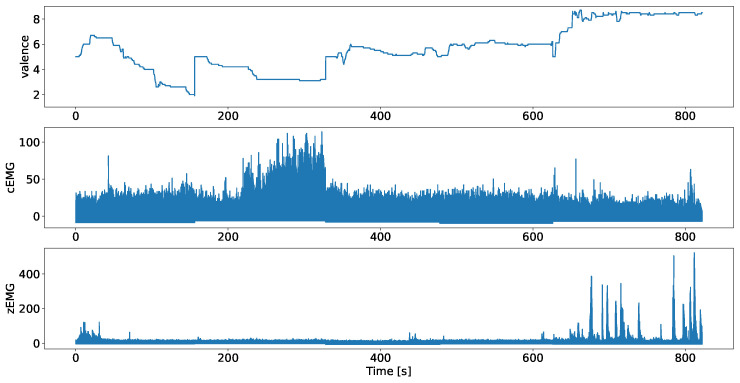
Valence ratings, and plots of the facial EMG data of corrugator supercilii (cEMG) and zygomatic major (zEMG), where data were obtained from separate studies, after preprocessing, using a participant example.

**Figure 2 sensors-24-01536-f002:**
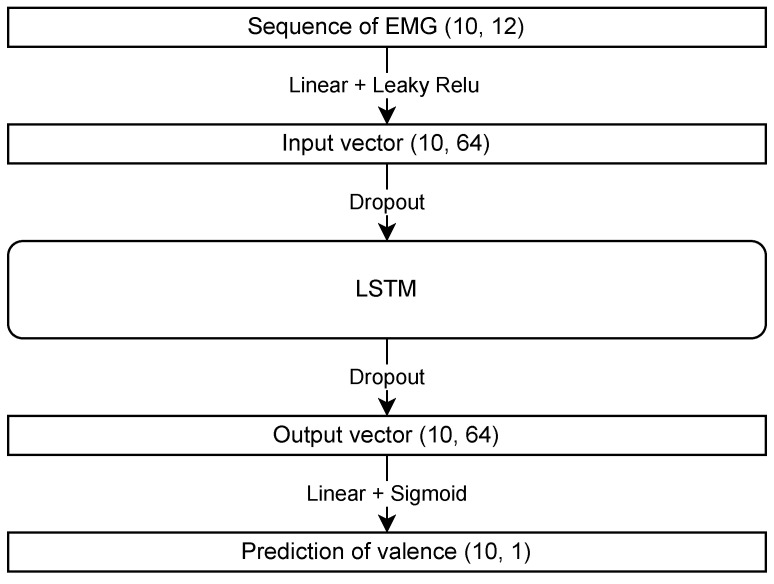
Model structure of long short-term memory (LSTM) showing each layer’s input and output types and activation functions.

**Figure 3 sensors-24-01536-f003:**
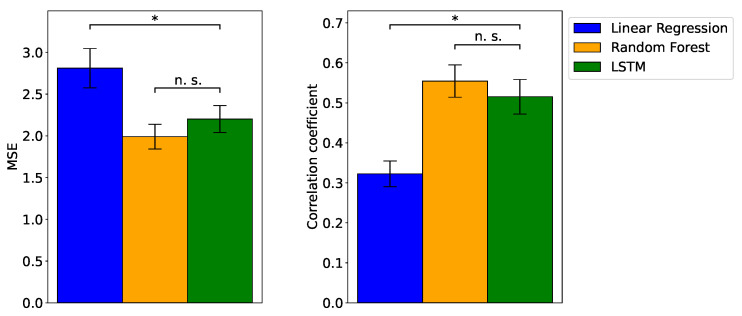
Mean, ± SE, mean squared error (MSE), and correlation coefficient for linear regression, random forest (RF), and long short-term memory (LSTM) models. The performance rankings, from the best to worst, are RF, LSTM, and linear regression. The comparison between the models indicates a significant difference (*) (p<0.05) between nonlinear machine learning (i.e., RF and LSTM) and linear regression models. There is no significant difference (*n.s.*) between machine learning models.

**Figure 4 sensors-24-01536-f004:**
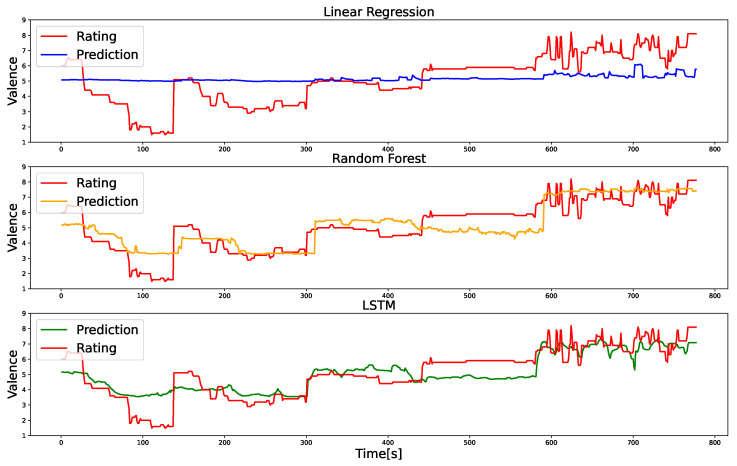
Representative examples of actual and predicted valence values obtained from linear regression, random forest (RF), and long short-term memory (LSTM) models. The red line corresponds to the labeled data, representing information from five films collected for a single participant. This figure highlights that RF and LSTM predict valence more accurately than linear regression.

**Figure 5 sensors-24-01536-f005:**
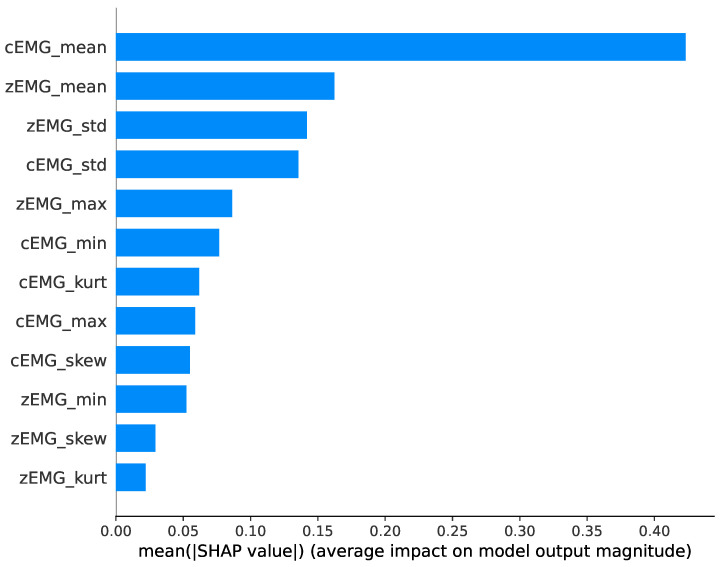
Average absolute SHapley Additive exPlanation (SHAP) values for each feature, showing the impact of each feature on valence prediction.

**Figure 6 sensors-24-01536-f006:**
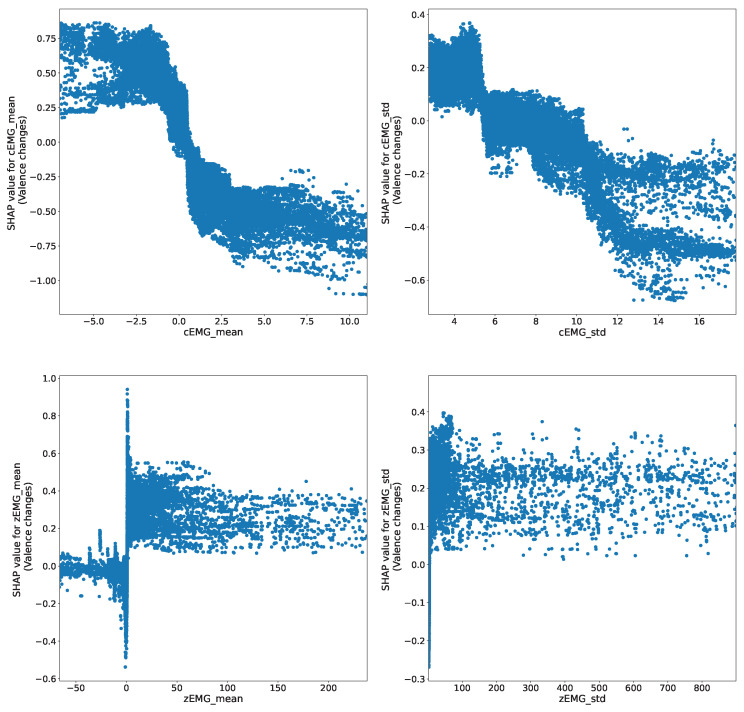
SHapley Additive exPlanation (SHAP) dependency plots show the representative relationships between predicted valence and facial electromyography (EMG) features. The predicted valence varies with the size of the input features.

**Figure 7 sensors-24-01536-f007:**
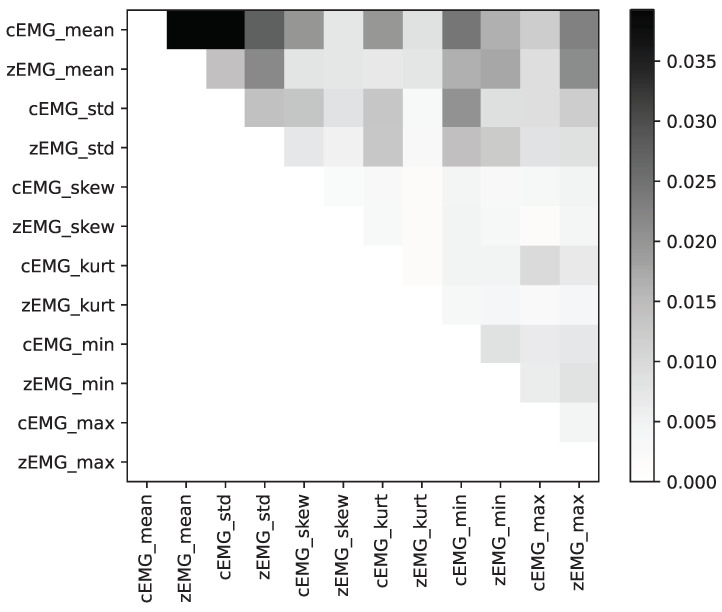
Average absolute SHapley Additive exPlanation (SHAP) interaction values for each feature, with darker colors indicating higher importance between features.

**Figure 8 sensors-24-01536-f008:**
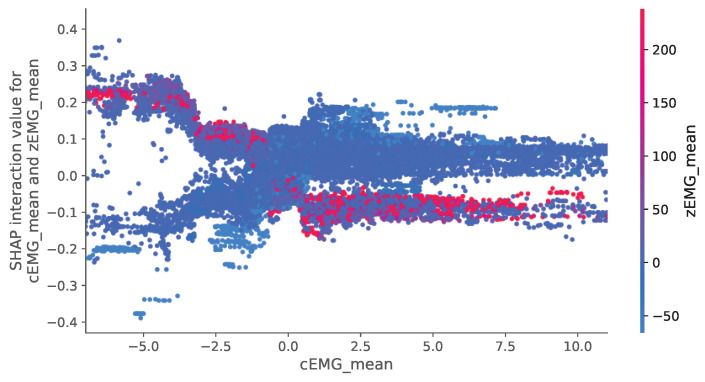
SHapley Additive exPlanation (SHAP) dependency plot demonstrating how valence prediction changes depend on the values between corrugator supercilia (cEMG) and zygomatic major (zEMG) features. A representative interaction between mean values is illustrated.

## Data Availability

The data supporting the findings of this study are available at https://dmsnc0.riken.jp/nextcloud/index.php/s/sKazk55PjtNtnRX.

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
