# Peer review of "Machine Learning-Based Interpretable Modeling for Subjective Emotional Dynamics Sensing Using Facial EMG"

_sensors, 2024, doi:10.3390/s24051536_

Round 1
Reviewer 1 Report
Comments and Suggestions for Authors
The article titled "Machine learning-based interpretable modeling for subjective emotional dynamics sensing using facial EMG" present a study that explores the relationship between subjective emotional experiences and physiological signals, specifically facial electromyography (EMG) signals. The study aims to understand how subjective emotional valence dynamics relate to facial EMG changes. It addresses the limitations of previous research that primarily focused on linear relationships and explores the potential of machine learning (ML) models for more accurate predictions. I would appreciate further clarifications and enhancements on the following aspects:
1. The statement mentions, “Recent psychophysiological studies have revealed that changes in facial electromyography (EMG) signals, recorded from corrugator supercilii (cEMG) and zygomatic major (zEMG) muscles, exhibit a linear relationship with subjective emotional valence dynamics [5–7].” The argument would be more convincing if it could cite studies from other scholars rather than only referencing its own research articles.
2. While polynomial or nonlinear models can fit regression models, i.e., analyzing EMG signal changes through nonlinear models to predict subjective emotional valence, does this imply a nonlinear relationship between EMG signal changes and subjective emotional valence? Please provide a list of relevant studies on nonlinear relationships.
3. Please clarify whether the statements "Nonlinear regression analyses have not been employed to investigate the association between subjective emotional valence dynamics and facial EMG changes" (lines 39-40) and "the relationship between subjective emotional valence dynamics and facial EMG changes is nonlinear" (lines 51-52) are equivalent concepts.
4. The statement in lines 52-54 mentions, "a previous study [12] reported that ML models outperformed linear analyses in classifying positive, neutral, and negative emotional states." Please clarify which algorithmic models are included in machine learning models. It is incorrect to exclude linear models from the definition of machine learning models.
5. Suggest changing "Section 2.1 Data " to " Section 2.1 Data acquisition."
6. In the statement "The performance rankings, from best to worst, are as follows: linear regression, RF, and LSTM," why wasn't LSTM included in the SHAP Analysis in Section 3.2? Please address this inconsistency in the analysis.
7. In the conclusion, it would be valuable to emphasize the need for further empirical research and practical experiments to validate the findings and overcome the limitations identified.
8. Please verify the correctness of the references, particularly reference on line 78 and line192.
Author Response
Dear Reviewer,
Thank you for your useful and constructive comments on our manuscript. We have carefully revised the manuscript according to your suggestions. Major changes to the manuscript are shown in red text. A professional English-language editing service (https://www.enago.jp/) made language-related changes, which are not highlighted unless the content was altered.
Q1.
The statement mentions, “Recent psychophysiological studies have revealed that changes in facial electromyography (EMG) signals, recorded from corrugator supercilii (cEMG) and zygomatic major (zEMG) muscles, exhibit a linear relationship with subjective emotional valence dynamics [5–7].” The argument would be more convincing if it could cite studies from other scholars rather than only referencing its own research articles.
Response
As suggested, we have referenced studies conducted by other researchers that report the association between facial EMG activity and dynamic valence ratings in the introduction (p.2, lines 37-43).
Q2.
While polynomial or nonlinear models can fit regression models, i.e., analyzing EMG signal changes through nonlinear models to predict subjective emotional valence, does this imply a nonlinear relationship between EMG signal changes and subjective emotional valence? Please provide a list of relevant studies on nonlinear relationships.
Response
Considering that some researchers have introduced a polynomial model as a simple approach for assessing nonlinear associations, we have referenced these articles in the introduction (p.2, lines 46-47). However, we did not find any additional relevant studies concerning nonlinear relationships between changes in the EMG signal and dynamic valence ratings.
Q3.
Please clarify whether the statements "Nonlinear regression analyses have not been employed to investigate the association between subjective emotional valence dynamics and facial EMG changes" (lines 39-40) and "the relationship between subjective emotional valence dynamics and facial EMG changes is nonlinear" (lines 51-52) are equivalent concepts.
Response
In the previous version of the manuscript, we attempted to delineate equivalent concepts. However, considering your suggestion, we acknowledge that the statement “nonlinear regression analyses have not been employed” posed a problem because polynomial regression was used to address this issue. Consequently, in the Introduction section (p.2, lines 46-49), we have replaced this sentence with:
“Although polynomial regression (a simple method to evaluate nonlinear associations [10,11]) was used in a previous study [7] and indicated the superior performance of quadratic models over linear ones, more complex nonlinear associations were not thoroughly examined.”
Q4.
The statement in lines 52-54 mentions, "a previous study [12] reported that ML models outperformed linear analyses in classifying positive, neutral, and negative emotional states." Please clarify which algorithmic models are included in machine learning models. It is incorrect to exclude linear models from the definition of machine learning models.
Response
We apologize for the error in our description. On further review, we realized that the previous study we cited did not compare ML and linear models. Consequently, we have omitted this reference in the revised version of the manuscript.
Q5.
Suggest changing "Section 2.1 Data " to " Section 2.1 Data acquisition."
Response
As advised, we have updated the subsection title to “Data Acquisition.”
Q6.
In the statement "The performance rankings, from best to worst, are as follows: linear regression, RF, and LSTM," why wasn't LSTM included in the SHAP Analysis in Section 3.2? Please address this inconsistency in the analysis.
Response
First, we acknowledge and apologize for the error in our initial description. The correct performance rankings, from the best to worst, are RF, LSTM, and linear regression (p. 7, figure 3).
We did not apply the SHAP analysis to LSTM. Some researchers have discussed that a SHAP analysis might pose challenges for sequence-dependent models such as LSTM [21]. This rationale is now explained in the introduction (p. 2, lines 68-69). Applying SHAP to LSTM results in a matrix of size “number of features” x “number of segments” for each instance. We are concerned that this complexity may draw attention from the primary focus of our research.
Q7.
In the conclusion, it would be valuable to emphasize the need for further empirical research and practical experiments to validate the findings and overcome the limitations identified.
Response
Our research is exclusively focused on analyzing subjective emotional changes during film viewing. To ascertain the universality of our findings, it is imperative to investigate various emotion-evoking methods. Furthermore, the SHAP analysis employed in our study is designed to explore correlations between features and predicted values and not to prove causation. Therefore, to confirm causal relationships, conducting empirical experiments is essential. We have incorporated this clarification into our manuscript (p. 14, lines 391-394).
Q8.
Please verify the correctness of the references, particularly reference on line 78 and line 192.
Response
We have corrected an error in the citation of references.
Reviewer 2 Report
Comments and Suggestions for Authors
The paper addresses an interesting and novel topic of using machine learning (ML) models to model the relationship between subjective emotional valence dynamics and facial electromyography (EMG) signals. The paper claims that ML models can better fit the nonlinear and interactive associations than conventional linear models, and provides empirical evidence from two previous studies.
The abstract is well-written and concise, but it could be improved by adding a sentence about the implications or applications of the findings. For example, how can the ML models be used to enhance emotion recognition or intervention systems?
The introduction should provide the background and motivation for the study. A literature review section may be added.
The materials and methods section is clear and detailed, but it could be more organized. The figure captions should also be more informative and explain what the figures show and why they are relevant.
The models section should explain the rationale and the advantages of using random forest (RF) and long short-term memory (LSTM) models for the task, and compare them with other possible ML models, such as support vector machines, neural networks, etc. The section should also provide the details of the model parameters, such as the number of trees, the depth, the learning rate, the activation function, etc.
The SHAP analysis section should explain what SHAP is and how it works, and cite the original paper that proposed it. The section should also interpret the results of the SHAP analysis and discuss the implications for the subjective–physiological relationship. For example, what do the nonlinear and interactive associations mean? How do they differ from the linear relationship? What are the limitations or assumptions of the SHAP analysis?
The conclusion should restate the main contributions, highlight the novelty and significance of the study, and provide some practical or theoretical implications or recommendations.
Overall, the paper has a lot of potential, but it needs minor revisions before it can be published. The paper should improve the organization and clarity of the other sections. The paper should also provide more justification and comparison for the choice of the ML models, and more interpretation and discussion for the results of the SHAP analysis. I hope that my comments are helpful and I look forward to seeing the revised version of the paper.
Comments on the Quality of English Language
grammatical errors of there in few occasions
Author Response
Dear Reviewer,
Thank you for your useful and constructive comments on our manuscript. We have carefully revised the manuscript according to your suggestions. Major changes to the manuscript are shown in red text. A professional English-language editing service (https://www.enago.jp/) made language-related changes, which are not highlighted unless the content was altered.
Q1.
The abstract is well-written and concise, but it could be improved by adding a sentence about the implications or applications of the findings. For example, how can the ML models be used to enhance emotion recognition or intervention systems?
Response
As suggested, we have corrected the abstract.
Q2.
The introduction should provide the background and motivation for the study. A literature review section may be added.
Response
As suggested, we have provided a more detailed background of the present study and clarified our motivation in the introduction (pp. 1–3). We have included additional information on previous studies related to this issue. Furthermore, we have elaborated on our motivation to demonstrate that interpretable ML approaches, compared with conventional ones, can enable a more accurate prediction of subjective emotional states—a practical utility. Additionally, we highlight our interest in gaining insights into the association between subjective and physiological concordance, which holds theoretical implications for the physiological mechanisms of emotion. Owing to the limited availability of direct evidence on this issue, we have refrained from creating a separate literature review section.
Q3.
The materials and methods section is clear and detailed, but it could be more organized. The figure captions should also be more informative and explain what the figures show and why they are relevant.
Response
Considering your suggestions, we have thoroughly revised and reorganized our manuscript’s “Materials and Methods” section. Moreover, to improve the clarity of the information conveyed in our figures, we have supplemented the captions with more detailed descriptions.
Q4.
The models section should explain the rationale and the advantages of using random forest (RF) and long short-term memory (LSTM) models for the task, and compare them with other possible ML models, such as support vector machines, neural networks, etc. The section should also provide the details of the model parameters, such as the number of trees, the depth, the learning rate, the activation function, etc.
Response
In response to your recommendations, we have introduced a section delineating the rationale and advantages of employing random forest (RF) and long short-term memory (LSTM) models (p. 4 164-166). Additionally, we clarify why support vector machines and basic neural networks were excluded from our study owing to their inadequate accuracy in preliminary experiments(p. 4 165-167). Furthermore, we have incorporated detailed information regarding the model parameters and types of activation functions used in the RF model, which we had not previously specified.
Q5.
The SHAP analysis section should explain what SHAP is and how it works, and cite the original paper that proposed it. The section should also interpret the results of the SHAP analysis and discuss the implications for the subjective–physiological relationship. For example, what do the nonlinear and interactive associations mean? How do they differ from the linear relationship? What are the limitations or assumptions of the SHAP analysis?
Response
Considering your observation, we have enriched the section on SHAP analysis by describing SHAP functionality and including citations(pp. 6-7, lines 242-245). The limitation of this form of analysis, as highlighted, is that it analyzes the relationship between input features and the model’s predicted values without demonstrating a causal relationship with actual subjective emotions. In acknowledging this, we emphasize the necessity to conduct empirical experiments to validate the psychological validity of the SHAP analysis. Furthermore, we have delved into the implications of our SHAP results in more detail in the discussion (p. 15).
Q6.
The conclusion should restate the main contributions, highlight the novelty and significance of the study, and provide some practical or theoretical implications or recommendations.
Response
In response to your suggestions, we have expanded the conclusions to include a description of the major contributions and the significance of this study. Additionally, we have incorporated examples of potential practical developments and outlined future perspectives. Furthermore, we have included recommendations for enhancing the understanding of emotions in psychology.
Reviewer 3 Report
Comments and Suggestions for Authors
Not an easy paper for the average clinician to read and understand. A lot of mathematical details. The authors should decide which audience they would like to address - this paper is certainly not for clinicians
Comments on the Quality of English LanguageAcademically reasonable but difficult to follow
Author Response
Dear Reviewer,
Thank you for your useful and constructive comments on our manuscript. We have carefully revised the manuscript according to your suggestions. Major changes to the manuscript are shown in red text. A professional English-language editing service (https://www.enago.jp/) made language-related changes, which are not highlighted unless the content was altered.
Q1
Not an easy paper for the average clinician to read and understand. A lot of mathematical details. The authors should decide which audience they would like to address - this paper is certainly not for clinicians
Response
As suggested, we have revised the descriptions in the introduction, methods, and discussion sections. We hope the revised manuscript is now more accessible and understandable for clinical researchers.